# RNAi-mediated rheostat for dynamic control of AAV-delivered transgenes

Megha Subramanian[1], James McIninch [1], Ivan Zlatev [1], Mark K. Schlegel [1], Charalambos Kaittanis [1], Tuyen Nguyen[1], Saket Agarwal [1], Timothy Racie[1], Martha Arbaiza Alvarado[1], Kelly Wassarman[1], Thomas S. Collins[1], Tyler Chickering[1], Christopher R. Brown[1], Karyn Schmidt[1], Adam B. Castoreno[1], Svetlana Shulga-Morskaya[1], Elena Stamenova[1], Kira Buckowing[1], Daniel Berman [1], Joseph D. Barry[1], Anna Bisbe[1], Martin A. Maier [1], Kevin Fitzgerald[1] & Vasant Jadhav [1] ✉

Adeno-associated virus (AAV)-based gene therapy could be facilitated by the development of molecular switches to control the magnitude and timing of expression of therapeutic transgenes. RNA interference (RNAi)-based approaches hold unique potential as a clinically proven modality to pharmacologically regulate AAV gene dosage in a sequence-specific manner. We present a generalizable RNAi-based rheostat wherein hepatocyte-directed AAV transgene expression is silenced using the clinically validated modality of chemically modified small interfering RNA (siRNA) conjugates or vectorized co-expression of short hairpin RNA (shRNA). For transgene induction, we employ REVERSIR technology, a synthetic high-affinity oligonucleotide complementary to the siRNA or shRNA guide strand to reverse RNAi activity and rapidly recover transgene expression. For potential clinical development, we report potent and specific siRNA sequences that may allow selective regulation of transgenes while minimizing unintended off-target effects. Our results establish a conceptual framework for RNAi-based regulatory switches with potential for infrequent dosing in clinical settings to dynamically modulate expression of virally-delivered gene therapies.

Adeno-associated virus (AAV) vectors are the leading platform for most in vivo gene therapy applications with potential as a years- if not life-long disease treatment option following a single dose[1]. However, one key challenge that has become evident from clinical trials of systemic AAV-mediated gene therapy is the wide individual-to-individual differences in therapeutic protein expression at the same vector dose. This could increase the risk of toxicity due to supraphysiological transgene expression in some cases[2,3]. Additionally, there remain significant difficulties in predicting the efficacious therapeutic dose range in humans from preclinical data. Together, these underscore the need for clinically translatable approaches to modulate transgene expression after AAV administration[4–6].

RNA interference (RNAi) is an evolutionarily conserved mechanism in which endogenous [microRNA (miRNA)] or exogenous siRNA/shRNA downregulate mRNA transcripts in a sequence-dependent manner[7]. As a native pathway that uses an efficient cellular catalytic mechanism, RNAi can achieve robust, durable, and specific silencing of gene transcripts of interest.

In recent years, several RNAi-based drugs have been successfully validated in clinical studies, with demonstrated benefit at low and infrequent doses. Novel delivery solutions along with highly chemically modified RNAs have improved potency, durability, and safety of RNAi therapeutics. This has culminated in five approved drugs and several others in clinical development[8–10]. In liver,

[1]Alnylam Pharmaceuticals, Cambridge, MA 02142, USA. ✉e-mail: vjadhav@alnylam.com

subcutaneous delivery of metabolically stabilized siRNA conjugated to N-acetylgalactosamine (GalNAc) results in potent silencing of gene expression that persists for months in humans with favorable safety and tolerability profiles[11–13]. Recent work has also broadened the scope for siRNA delivery to extrahepatic tissues, with conjugation of 2′-O-palmityl (C16) demonstrating wide distribution and durable target knockdown across cell types in central nervous system, eye, and lung[14]. All of these advances for silencing of endogenous disease-associated genes via RNAi also hold the potential for on-demand regulation of exogenously-delivered transgenes in a therapeutic setting.

The small size of siRNA or shRNA binding sites (typically 19-23 nucleotides) enables them to be readily incorporated within the limited packaging capacity of viral genomes. Incorporating binding sites for tissue-specific miRNAs has been exploited to improve the specificity of AAV transgene targeting by selectively reducing expression in undesired cell types[15–17].

Prior designs for RNAi-based on-switches have used exogenously-delivered molecules to control the processing of shRNAs delivered alongside the therapeutic transgene or to control accessibility of endogenous microRNAs to artificially-inserted binding sites on the viral mRNA[18–23]. However, their application has been limited due to either low dynamic range for modulation, off-target risks, or a lack of validation in the clinical setting[18]. In contrast, supplying chemically stabilized siRNAs exogenously overcomes the reliance on endogenous miRNAs and offers precise and flexible control of dosage.

As an alternative to siRNAs that require repeated, though infrequent administration, RNAi via shRNAs that can be stably introduced into AAV vectors may allow continuous regulation of the expressed transgene in cis as a single treatment[24–27]. Previous studies have shown that shRNAs can be expressed at high levels alongside therapeutic transgenes in a single AAV vector as a strategy for long-term gene complementation to treat toxic gain-of-function diseases[23,28].

While exogenous RNAi modalities may enable low basal expression of transgenes, the versatility of these systems would be improved if transgene silencing can be reversed to control expression in the on-state. We have previously reported a highly potent and generalizable approach for in vivo control of RNAi pharmacology using a short, synthetic single-stranded oligonucleotide known as REVERSIR[29]. They functionally abrogate RNAi activity by acting as synthetic high-affinity decoys to sequester RNA-induced silencing complexes (RISC) loaded with complementary siRNA antisense (guide) strands. REVERSIR binding thereby prevents RISC-mediated recognition and degradation of target mRNA transcripts and consequently allows their translation. The development of REVERSIR as an antidote for RNAi activity represents a valuable tool that may be co-opted to exogenously regulate the on-state of transgenes that are under RNAi control. The potential use of REVERSIR in therapeutic settings is supported by early clinical studies showing human translation of synthetic single-stranded oligonucleotides that block the activity of RISC complexes loaded with disease-associated endogenous microRNAs (anti-miR technology)[30–34].

Here, we investigate the potential utility of combining RNAi-mediated knockdown with REVERSIR-enabled rescue of transgene expression as a rheostat for AAV-delivered genes. Using hepatotropic recombinant AAV8 (AAV8) vectors as a model, we assessed both exogenous siRNA delivery and viral co-expression of shRNA alongside the transgene as approaches for silencing transgene expression. We demonstrate efficient upregulation of transgene levels by blocking activity of siRNA/shRNA with REVERSIR technology. Whereas the standard high-stability REVERSIR design enabled long-term transgene induction, a less metabolically stable chemical template allowed transient transgene upregulation and consequent resumption of RNAi-mediated silencing.

Finally, we describe novel and highly specific siRNA sequences that may be employed for clinical development to modulate expression from exogenous vector delivery systems. These siRNA sequences exhibited robust RNAi activity without causing undesired off-target gene silencing within the endogenous transcriptome of humans and mammalian preclinical models. Together, our studies support the development of RNAi and related REVERSIR technology for regulation of AAV transgenes, particularly for temporal control and dosage refinement in the event of highly variable transduction efficiency in the clinical setting.

## Results

### REVERSIR-mediated induction of transgene under control of vectorized shRNA

We first evaluated potential of a single agent approach with silencing of transgene expression by vectorized shRNA in cis and on-demand induction by REVERSIR (Fig. 1a). To explore this, we initially examined the ability of REVERSIR to rescue shRNA-mediated silencing using luciferase assays in cultured cells. We assessed in trans repression by co-transfecting transthyretin (Ttr)-targeting shRNA or control non-targeting (NT) shRNA embedded within an optimized human miR-30 (miR-30E) or a mouse miR-33 (Supplementary Data 1) primary micro-RNA (pri-miRNA) scaffold along with a luciferase vector containing a fully complementary Ttr target element within the 3′ UTR[24,26]. Expression of TTR shRNA from both scaffolds resulted in significant suppression of luciferase reporter activity relative to the NT shRNA. In both cases, dose-dependent recovery of luciferase reporter expression was observed with co-transfection of 22-mer TTR REVERSIR but not with a control non-targeting REVERSIR (NT REVERSIR) (Supplementary Table 1) of the same length and chemistry (Supplementary Fig. 1a and Supplementary Fig. 1b).

Next, we generated a single self-regulating AAV vector that would concurrently express a regulatory shRNA along with a transgene mRNA harboring a cognate 3′ UTR target site for the shRNA (Fig. 1a). To achieve this, we inserted the miRNA-embedded TTR shRNA cassette into a chimeric intron residing between a Gaussia luciferase (GLuc) reporter transgene and liver-specific TBG promoter and also included a single fully matched binding site for the TTR shRNA within the 3′ UTR of GLuc (Fig. 1b and Supplementary Data 1). This configuration allows expression of the regulatory shRNA and transgene mRNA to be coupled within a single transcriptional unit but permits each to be processed independently following precursor-mRNA splicing (Fig. 1a).

In vitro evaluation of self-silencing AAV plasmids demonstrated robust basal downregulation of the secreted GLuc reporter transgene when both a miRNA-embedded TTR shRNA and its complementary target site (TTR-ts) were present in the vector, relative to matched constructs expressing a NT shRNA or scrambled target site (NT-ts) (Fig. 1c and Supplementary Fig. 1c). Consistent with the need for miRNA-embedded shRNA to undergo multiple intracellular processing steps prior to generating active RISC effectors, longitudinal monitoring of transgene expression in vitro showed a delayed onset of GLuc suppression. Reporter levels were comparable for both self- and non-self-regulating designs at early timepoints with significant silencing only evident beginning at 14 h post-transfection (Fig. 1c).

Co-transfection of the AAV vectors with a 22-mer REVERSIR directed against the TTR shRNA resulted in recovery of GLuc protein and mRNA expression (Fig. 1d and Supplementary Fig. 1d). Consistent with a sequence-specific mechanism, we did not observe reversal of shRNA-induced knockdown of GLuc with NT REVERSIR (Fig. 1d and Supplementary Fig. 1d).

We next assessed in vivo feasibility of this strategy using vectors expressing miR-33-embedded shRNA cassettes since prior studies had shown that this pri-miRNA backbone retained high efficacy without compromising AAV genome integrity during viral replication[24,25]. In line with our in vitro findings, expression of the self-silencing AAV containing both an intronic miR-33 TTR shRNA and its complementary binding site (TTR shRNA/TTR-ts) led to a sustained knockdown (~76%)

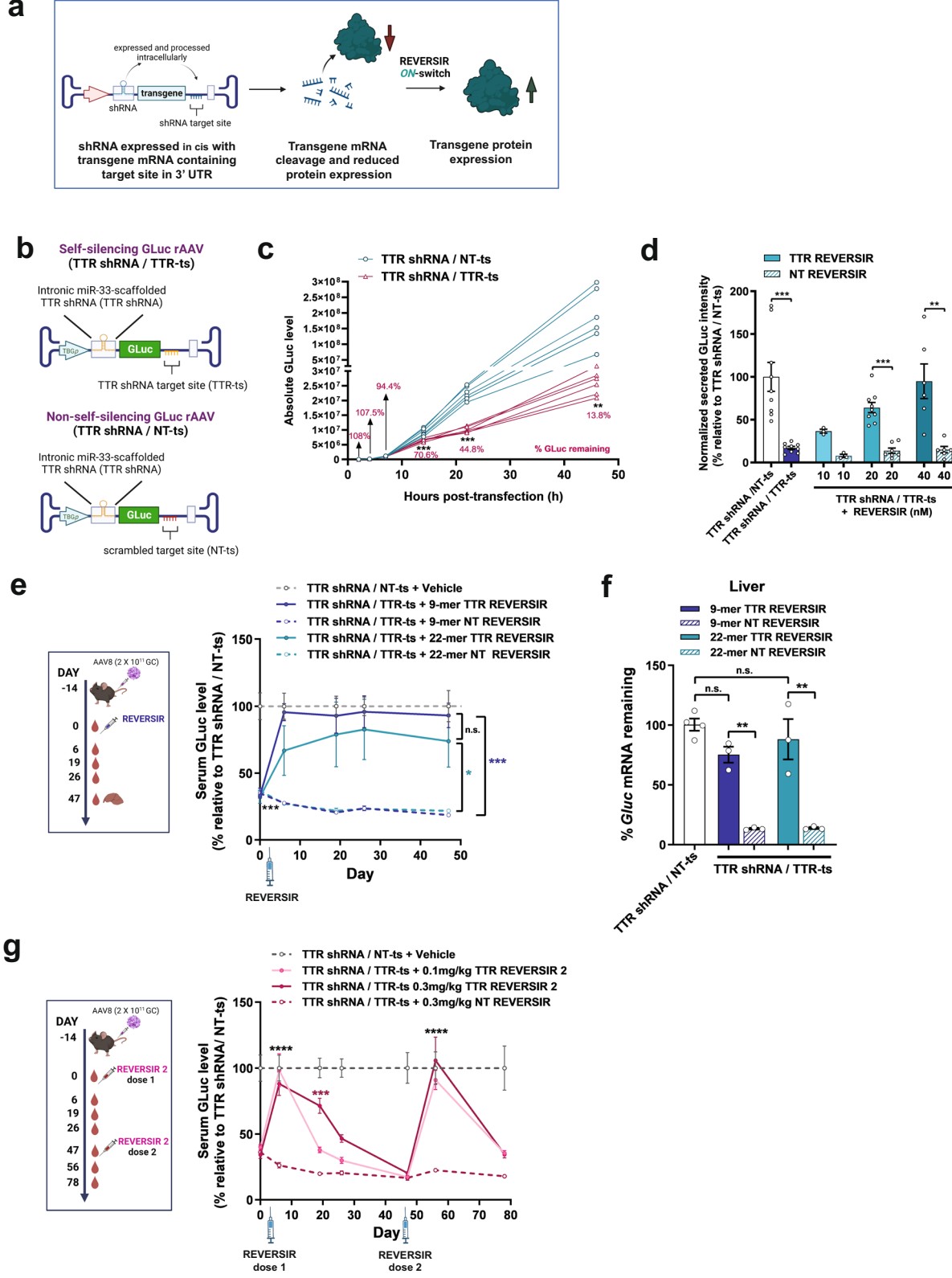

of secreted GLuc reporter signal, relative to the non-self-silencing (TTR shRNA/NT-ts) vector design (Fig. 1e).

GLuc serum levels recovered and were comparable to the control non-self-silencing vector within 7 days following molar equivalent subcutaneous (SC) dosing of either a short 9-mer or long 22-mer TTR REVERSIR, with induction lasting for at least 6 weeks. As expected, no significant GLuc induction was observed upon treatment with

equivalent doses of the control NT REVERSIR. Consistently, qRT-PCR analyses also revealed rescue of *Gluc* transcript levels in the presence of TTR REVERSIR relative to the NT REVERSIR control (Fig. 1f). These data successfully establish induction of self-silenced transgene using REVERSIR approach.

For more transient upregulation of transgene expression, we hypothesized that reducing the metabolic stability of the REVERSIR

**Fig. 1 | Transgene induction from shRNA-regulated self-silencing AAV vector using REVERSIR. a** Diagram illustrating AAV transgene self-silencing using intronically-encoded shRNA and induction of expression with exogenously-administered REVERSIR. **b** Viral genome schematics of AAV8 GLuc reporter vector with intronic TTR shRNA and complementary (TTR shRNA/TTR-ts) or scrambled 3' UTR target site (TTR shRNA/NT-ts) **c** HepG2 cells were transfected with AAV plasmids and media was collected at each timepoint. Media was fully exchanged at every collection, with each line corresponding to accumulated GLuc in supernatant from a single well since prior timepoint ($n = 6$ wells per condition; two-way repeated measures ANOVA with Sidak's correction). **d** Recovery of reporter expression with REVERSIR in HepG2 cells. AAV constructs in (**b**) were co-transfected with indicated concentrations of 22-mer TTR REVERSIR or NT REVERSIR, along with a Luc2 internal control plasmid. 48 h post-transfection, GLuc and Luc2 intensities were assayed in cell supernatant and lysate, respectively. GLuc/Luc2 ratios were computed and plotted relative to TTR shRNA/NT-ts condition set to 100%. Data were

analyzed by one way ANOVA with Sidak's correction. (**e–g**) Mice were injected with $2 \times 10^{11}$ genome copies (GC) of AAV vectors in (**b**) 2 weeks before molar equivalent dosing of 9-mer (0.1 mg/kg) or 22-mer (0.2 mg/kg) TTR or NT REVERSIR on Day 0 (D0). **e** Serum GLuc levels at indicated timepoints prior to and following REVERSIR treatment ($n = 5$ animals in PBS group; $n = 3$ animals for all other groups). Statistical analysis was performed using two-way repeated measures ANOVA followed by Tukey's test. **f** qRT-PCR for *Gluc* transcript levels in liver tissue at terminal timepoint normalized to endogenous *Gapdh* control. Data were analyzed by one-way ANOVA with Tukey's post hoc test. **g** Serum GLuc levels in mice administered with 0.1 mg/kg or 0.3 mg/kg 9-mer TTR REVERSIR 2 or NT REVERSIR on D0, followed by a second dose on D47 ($n = 3$). Data from the same TTR shRNA/NT-ts + vehicle group is shown in (**e**) and (**g**). All error bars represent s.e.m. $^{*}p < 0.05$ $^{**}p < 0.01$ $^{***}p < 0.001$ $^{****}p < 0.0001$. n.s., not significant as determined by indicated post-hoc test. Graphics were created using BioRender.com. Source data and statistics are provided in the Source Data file.

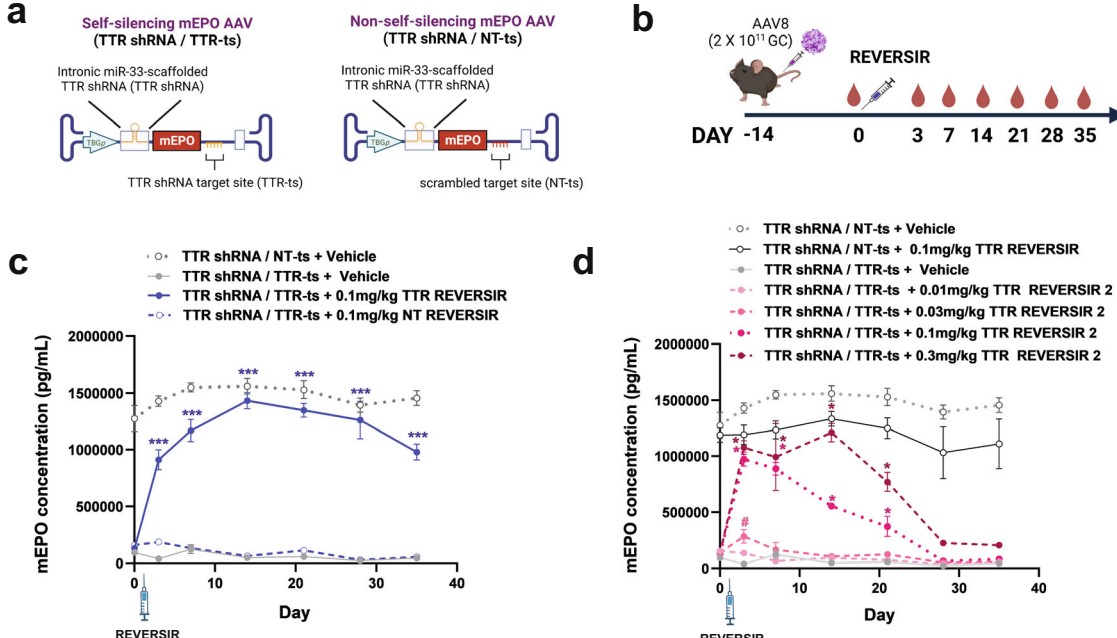

**Fig. 2 | REVERSIR-mediated regulation of an erythropoietin transgene. a** Vector genome schematics for AAV encoding mouse EPO (mEPO) transgene under control of TTR shRNA with intact (self-silencing; TTR shRNA/TTR-ts) or scrambled (non-self-silencing; TTR shRNA/NT-ts) binding site in 3' UTR. **b** Mice were transduced with $2 \times 10^{11}$ GC of AAV8 vectors shown in (**a**) for two weeks and then subcutaneously injected with REVERSIR. Serum was collected for quantification of mEPO levels by ELISA at indicated days prior to and following REVERSIR administration. **c** Serum EPO concentrations in mice that received 0.1 mg/kg 9-mer TTR or NT REVERSIR, compared to vehicle control on D0 ($n = 4$ mice for vehicle groups and $n = 3$ mice for REVERSIR groups). Data were analyzed by two-way repeated measures ANOVA with Tukey's post hoc test ($^{***}p < 0.001$ relative to TTR shRNA/TTR-ts + NT REVERSIR control). **d** EPO concentrations following treatment with increasing doses of TTR REVERSIR 2 (0.01, 0.03, 0.1, or 0.3 mg/kg). Experiments in

panels (**c**) and (**d**) were conducted in parallel and compared to the same vehicle conditions. An additional control was included in (**d**) in which mice transduced with the non-self-silencing vector were treated with TTR REVERSIR ($n = 4$ mice for vehicle and 0.03 mg/kg REVERSIR groups; $n = 3$ mice for 0.01 and 0.1 mg/kg REVERSIR groups; $n = 2$ mice for 0.1 and 0.3 mg/kg REVERSIR groups). Statistical testing was performed by using two-way repeated measures ANOVA followed by Tukey's test for multiple comparisons ($^{*}p < 0.05$ relative to TTR shRNA/TTR-ts + vehicle control). #, adjusted $p = 0.049$ at D3 compared to TTR shRNA/TTR-ts + vehicle condition by multiple unpaired *t*-test (one per row) corrected for multiple comparisons by Holm-Sidak method (alpha=0.05). All error bars represent s.e.m. AAV vector and experimental design illustrations were created with BioRender.com. Source data with detailed statistical analyses are provided as a Source Data file.

may lead to its more rapid intracellular clearance, thereby facilitating resumption of RISC-mediated target engagement and silencing. Towards this end, we dosed mice transduced with self-silencing TTR shRNA/TTR-ts AAV with a 9-mer TTR REVERSIR designed with lower LNA and PS content (TTR REVERSIR 2) to diminish metabolic stability[8,35].

TTR REVERSIR 2 led to initial recovery of GLuc transgene expression back to control levels by D6. Consistent with reduced metabolic stability, dose-dependent resumption of GLuc silencing was observed over time (Fig. 1g). A second peak in GLuc levels was observed following a re-challenge with another dose of TTR REVERSIR

2 on D47, highlighting the potential for repeated induction of transgene expression with tunable REVERSIR designs.

### In vivo control of an shRNA-regulated erythropoietin transgene with REVERSIR

To demonstrate that a single agent strategy using REVERSIR can be used to control physiologically-relevant transgenes, we designed a TTR shRNA-embedded self-regulating AAV vector encoding mouse erythropoietin (m*Epo*) (Fig. 2a, b). Even though AAV was administered at a high dose of $2 \times 10^{11}$ GC, we observed >92% suppression of serum mEPO levels in mice injected with AAV expressing TTR shRNA along

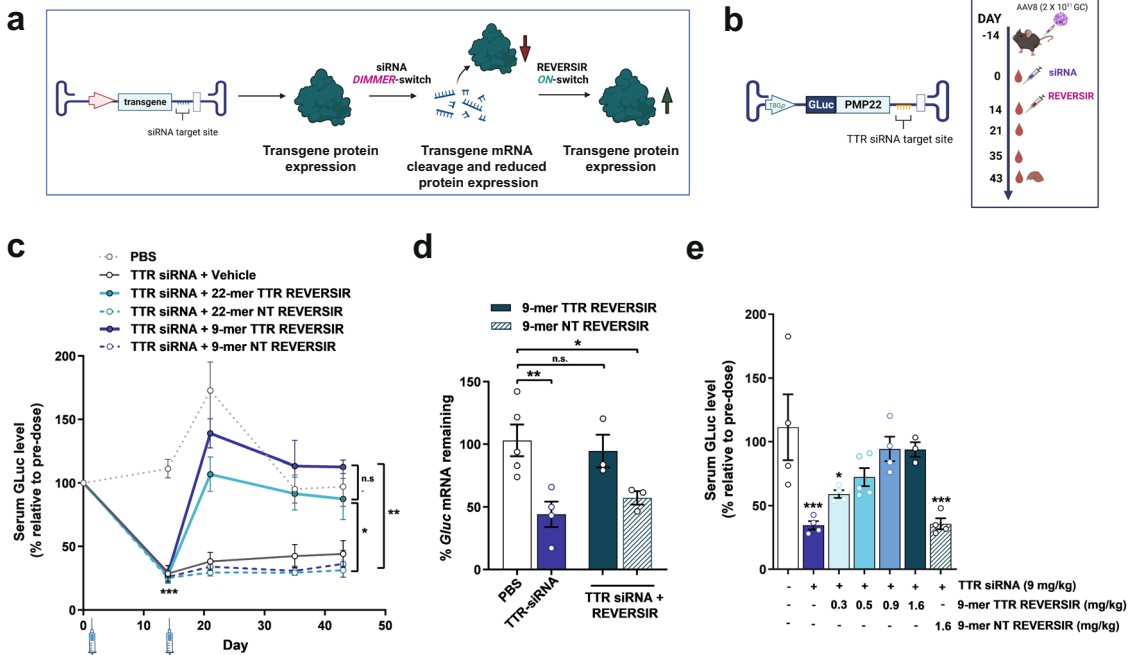

**Fig. 3 | In vivo regulation of an AAV-expressed reporter transgene by exogenous delivery of siRNA and REVERSIR. a** Diagram depicting dual agent approach involving suppression of AAV transgene expression with siRNA and subsequent upregulation of protein expression by abrogation of siRNA activity with REVERSIR. **b** Mice were injected with $2 \times 10^{11}$ GC of AAV encoding bicistronic expression of a GLuc reporter with a TTR siRNA binding site in the 3' UTR. Mice were subcutaneously injected with vehicle or TTR siRNA at 9 mg/kg (D0). This was followed on D14 with a single molar equivalent dose of full-length 22-mer (3 mg/kg) or 9-mer TTR REVERSIR (1.6 mg/kg) and compared to NT REVERSIR control. **c** Serum GLuc levels normalized to pre-dose (D0) for each animal ($n = 5$ for PBS; $n = 4$ for vehicle and 22-mer TTR REVERSIR groups; $n = 3$ for 22-mer NT REVERSIR and both 9-mer

REVERSIR groups). Data were analyzed by two-way repeated measures ANOVA followed by Tukey's post hoc test. **d** *Gluc* transcript levels in liver tissue at D42 normalized to *Gapdh* control. **e** Serum GLuc levels at D21 in AAV-transduced mice that received TTR siRNA (9 mg/kg; D0), followed by increasing doses of 9-mer TTR REVERSIR (D14) or NT REVERSIR ($n = 4$ for PBS, TTR siRNA, 0.3 mg/kg REVERSIR, and NT REVERSIR groups; $n = 5$ for 0.5 mg/kg REVERSIR group; $n = 3$ for 1.6 mg/kg TTR REVERSIR group). Data were analyzed by one-way ANOVA followed by Holm-Sidak or Dunnett's tests, respectively for (**d**) and (**e**). $^{*}p < 0.05$ $^{**}p < 0.01$ $^{***}p < 0.001$ n.s., not significant. Error bars represent s.e.m. Source data are provided as a Source Data file. Diagrams were created with BioRender.com.

with its cognate binding site relative to those harboring a non-targeted binding site. A single administration of TTR REVERSIR but not NT REVERSIR resulted in complete recovery of mEPO expression that lasted for at least 28 days (Fig. 2c). In contrast, treatment with a less stable REVERSIR design led to transient and dose-dependent increases in mEPO concentrations (Fig. 2d).

Results from these studies suggest a viable single-agent strategy involving constitutive self-suppression of the AAV transgene that may be counteracted by exogenous administration of REVERSIR to induce expression transiently or for a prolonged period of time.

**Transgene induction by REVERSIR blockade of exogenous siRNA activity**

Using hepatotropic AAV8, we sought to probe whether exogenous administration of GalNAc-conjugated siRNA and REVERSIR may enable on-demand modulation of AAV-expressed transgenes (Fig. 3a). Mice were injected with an AAV8 vector encoding a bicistronic transcript composed of a GLuc reporter and a second transcript (peripheral myelin protein 22, PMP-22), with a binding site for a TTR-targeting siRNA inserted in the 3' UTR (Supplementary Table 2 and Fig. 3b).

Single dose of a GalNAc-TTR siRNA resulted in sustained dose-dependent reduction of AAV-driven serum GLuc expression, with the highest dose of 9 mg/kg showing 71% silencing at 14 days (Fig. 3c and Supplementary Fig. 2a). Two weeks following siRNA treatment, mice received a single molar equivalent dose of TTR or NT REVERSIR (Fig. 3b; right). Both the 22-mer and 9-mer TTR REVERSIR molecules efficiently counteracted silencing activity by the 9 mg/kg siRNA dose, leading to prolonged and comparable induction of GLuc expression to

levels similar to that of vehicle-treated controls for at least 4 weeks (Fig. 3c and Supplementary Fig. 2b). In contrast, GLuc reporter knockdown was unaffected in the presence of NT REVERSIR. The qRT-PCR analyses also showed that treatment with GalNAc-TTR siRNA resulted in significant reductions in mRNA levels of *Gluc*, which were restored to baseline following administration of 9-mer TTR REVERSIR but not the control NT REVERSIR (Fig. 3d).

To additionally illustrate that induction may be titrated to achieve varying levels of protein expression, we monitored GLuc following dosing of siRNA-treated mice with increasing amounts of 9-mer TTR REVERSIR. We observed dose-dependent upregulation of GLuc levels one-week post-REVERSIR dosing, suggesting that REVERSIR can allow fine-tuning of AAV transgene expression to the desired level by modulating the magnitude of siRNA-mediated silencing activity (Fig. 3e).

To demonstrate that exogenous dosing of siRNA with REVERSIR represents a generalizable approach for off- and on-state control of AAV transgenes, we evaluated two additional AAV vectors in which we varied either the identity of the expressed transcript and/or the regulatory siRNA sequence. Mice were transduced with an AAV8 vector conferring expression of human Angiopoietin-like 3 (hANGPTL3) with a unique 3' UTR target site for a GLuc siRNA (Supplementary Table 2 and Supplementary Data 1). Activity of the GLuc siRNA and REVERSIR were confirmed in vitro using dual luciferase reporter assays (Supplementary Fig. 2c). In vitro, the 9-mer GLuc REVERSIR displayed lower potency compared to the 22-mer REVERSIR, recapitulating prior studies showing poorer performance of shorter REVERSIR molecules when delivered by lipid-based transfection (Supplementary Table 1 and Supplementary Fig. 2d)[29]. In vivo, treatment of AAV-transduced

mice with 9 mg/kg GLuc siRNA resulted in robust reduction of serum hANGPTL3 protein. Administration of 9-mer GLuc REVERSIR, but not NT REVERSIR, reverted serum hANGPTL3 to levels indistinguishable from vehicle controls with protein upregulation lasting for 3 weeks post-induction (Supplementary Fig. 2e).

We also tested an AAV construct encoding bicistronic expression of GLuc reporter with a 3′ UTR TTR siRNA target site as described previously. However, in this instance, we replaced the coding sequence of PMP-22 with that of a different gene, human Factor XII (hF12). Durable and significant reduction in GLuc levels of >60% were detected up to D49. Mice that received the TTR REVERSIR displayed significant upregulation of reporter expression to levels of vehicle-treated controls through D63 (Supplementary Fig. 2f).

Collectively, these results indicate that exogenous delivery of GalNAc-conjugated siRNAs and REVERSIR represents a generalizable strategy for titratable and on-demand control of transgene expression following AAV transduction.

### Identification of potent and specific transgene regulator siRNAs

Clinical application of an RNAi-based regulatory switch for AAV vectors necessitates highly selective and unique siRNA sequences that allow RNAi activity to be exclusively directed towards the intended exogenously-expressed transgene and no off-target effects on the endogenous transcriptome. Using bioinformatic approaches, we identified three candidate transgene regulator siRNAs (TR-siRNAs) that were designed to have little to no sequence complementarity to any annotated transcripts in human, cynomolgus macaques, rat, and mouse (Supplementary Table 2).

In the dual-reporter luciferase system, all three siRNAs exhibited significant dose-dependent on-target repression of luciferase activity, with IC$_{50}$ values in the low nanomolar range (Fig. 4a and Supplementary Data 1). To evaluate the potential for microRNA-like off-target knockdown, we co-transfected the siRNAs with luciferase sensors containing either a single or four tandem repeats of a site complementary to the seed region (nucleotides 2–8) of the siRNA antisense strand (Supplementary Data 1). No significant microRNA-like off-target luciferase reporter repression was observed at all doses for siRNAs 1 and 2. siRNA 3 showed significant knockdown at doses ≥10 nM with the off-target reporter containing 4 sites, but no off-target activity was observed at any dose for the reporter harboring a single seed-matched site (Fig. 4a).

To characterize the potential for sequence-dependent and -independent off-target effects more broadly, we transfected these candidate TR-siRNAs into Hep3B cells and primary murine hepatocytes to assess their global impact on the human and mouse transcriptomes by RNA-seq. As designed, there were relatively few transcripts that contained canonical seed matches (8mer, 7mer-m8, and 7mer-A1) to the TR-siRNA antisense strands within their 3′UTRs. Across both cell types, all three TR-siRNAs generally displayed minimal to no transcriptional dysregulation when comparing log$_2$ fold-changes in mRNA expression levels for genes harboring canonical seed-matched sites as well as those without canonical seed matches. Additionally, 2 of the 3 siRNAs showed no transcripts with a magnitude of dysregulation exceeding 2-fold (Fig. 4b and Supplementary Table 3). A *Tmprss6*-targeting siRNA that was assayed in parallel showed robust on-target transcript knockdown, confirming successful transfection of all the evaluated siRNAs.

To confirm that the observed lack of off-target gene dysregulation translates to a favorable safety profile in vivo, we administered the three TR-siRNAs at toxicological doses in rats. Rats received once weekly subcutaneous injections (qw x 3 doses) of 30 or 100 mg/kg siRNA, representing 2-3 log increase from the pharmacological dose range. All three siRNAs showed no significant liver enzyme elevations (Fig. 4c and Supplementary Fig. 3a). Consistent with our in vitro findings, RNA-seq evaluation of rat livers subjected to the toxicological

dose regimen indicated minimal evidence of transcriptional dysregulation among canonical 7-mer and 8-mer seed-matched transcripts, indicating a substantial lack of seed-mediated off-target gene signatures (Supplementary Fig. 3b).

Overall results from our in vitro RNA-seq and rat toxicity screening efforts establish these siRNAs as potent RNAi triggers with high on-target specificity. These TR-siRNA sequences display minimal propensity for off-target gene disruption and in vivo toxicity, lending initial support for their use as regulators of exogenous transgenes with potentially favorable safety profiles at least in the context of liver-directed applications.

## Discussion

A lack of broadly applicable methods to control the dosage of therapeutic transgenes is a central challenge surrounding AAV gene therapies, since most currently rely on persistent constitutive expression following a single administration. The ability to refine transgene dosage to within the targeted therapeutic range or suppress expression in case of adverse events represent important features that could maximize the safety of AAV-based therapies.

RNA-based molecular switches are attractive candidates to regulate transgene expression from AAVs owing to their small size, sequence specificity, and reduced potential for immunogenicity[18]. Recent developments in riboswitches based on steric oligo blockade of self-cleaving ribozyme activity and small molecule regulation of alternative RNA splicing have shown promise in preclinical models but their applicability in clinical settings is unknown[36,37]. We report a generalizable and clinically viable approach for dosage control of AAV-delivered cargos involving dynamic, robust, and reversible control via RNAi with benefit of infrequent dosing. We demonstrate reversal of RNAi-mediated transgene knockdown with REVERSIR in both a dual agent setting involving exogenous administration of chemically modified siRNA, and a single agent setting where co-delivery of vectorized shRNA enables constitutive silencing. Both approaches provide durable dampening of transgene expression that may be abrogated with REVERSIR to yield weeks-long elevation in protein levels in mice following a single subcutaneous dose of the inducer.

Unlike riboswitch designs that depend on self-cleaving ribozyme activity or splicing-dependent transgene cassettes, both of which afford extremely tight control in the off-state, an RNAi mechanism is unlikely to completely abolish basal transgene expression. Nevertheless, clinical trials evaluating investigational RNAi therapeutics have shown robust and prolonged target knockdown; results from the ORION Phase 3 study of a GalNAc-siRNA targeting PCSK9 support a once-every-6-month dosing regimen[38]. We showed dose-dependent lowering of AAV-expressed protein levels that were maintained for over a month in rodents after a single dose of siRNA or were constitutively suppressed in the case of vectorized shRNA.

Despite the slower kinetics of onset and inability to wholly inactivate transgene expression, there are certain settings where the durability of RNAi activity and the potential for infrequent dosing could have distinct utility for the regulation of AAV gene therapies. For instance, RNAi may be well suited for regulation of certain cargoes, such as monoclonal antibodies, where fine-tuning antibody concentrations below the required threshold for therapeutic effect may be sufficient to minimize adverse effects[39]. One could also consider using RNAi to address the challenge of dose scaling and management for highly active transgenes, such as the Padua variant of FIX, where small differences in vector dose could lead to large changes in protein production[6]. Systemic administration can also produce variable levels of transgene expression in response to the same vector dose, consequently increasing the risk of adverse toxicity if protein expression far exceeds the therapeutic range. This was recently highlighted in trials evaluating AAV-FVIII for the treatment of hemophilia A where a high degree of variation in FVIII activity levels was observed among

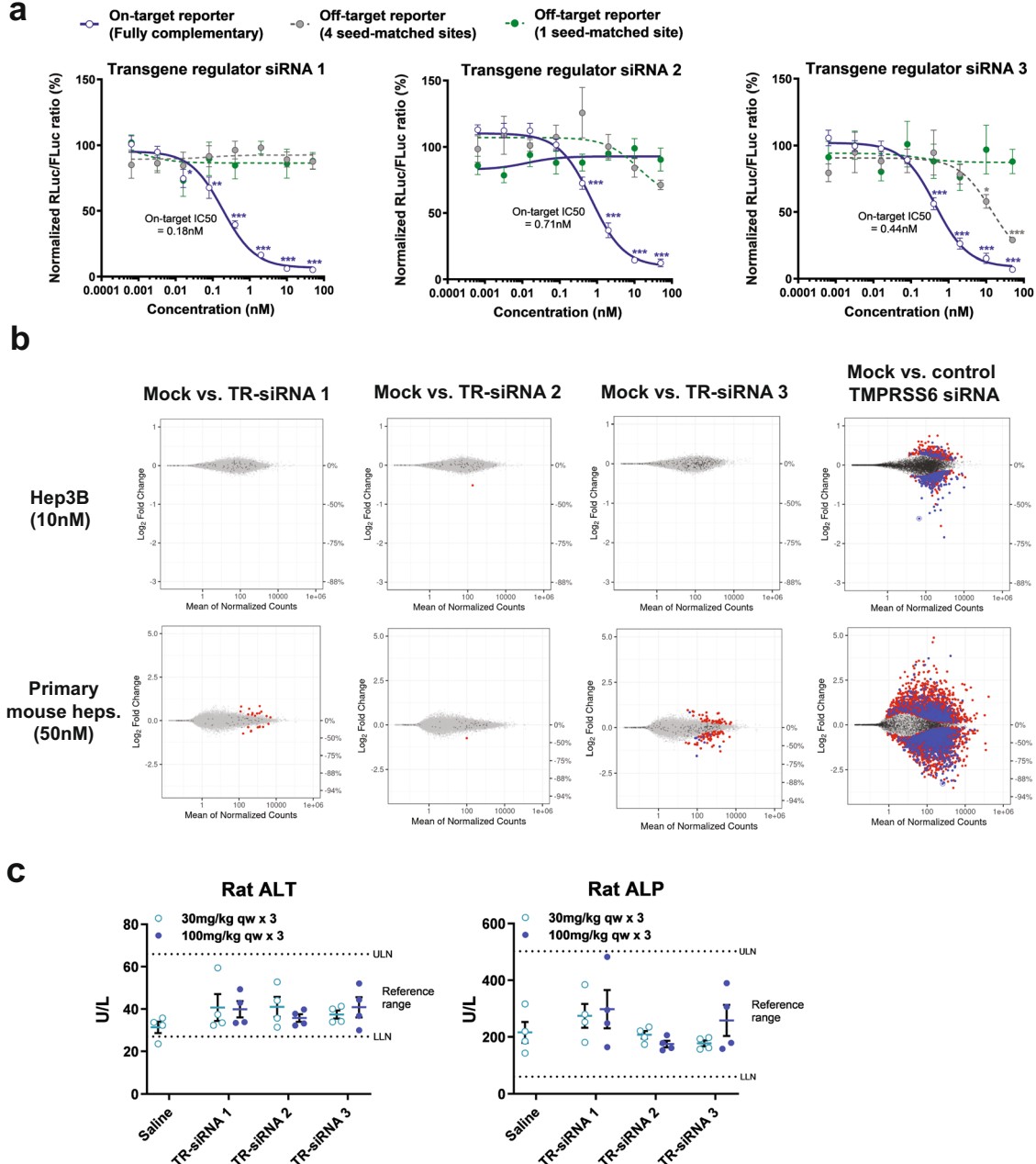

**Fig. 4 | In vitro characterization of on- and off-target activity of transgene regulator siRNA sequences. a** On-target silencing efficacy (solid blue line) of three lead transgene regulator siRNA (TR-siRNA) sequences as assayed by co-transfection of serially titrated doses of siRNA with dual luciferase sensors containing a perfectly matched binding site in the 3' UTR of Renilla luciferase (RLuc). Seed-mediated off-target repression was similarly assessed by dose-response activity of siRNA in the presence of luciferase reporters bearing either 1 (green dashed line) or 4 tandem (gray dashed line) seed-matched target sites. RLuc/FLuc ratios were normalized to the mock-transfected control (no siRNA) condition set at 100% and plotted as mean ±s.e.m. Data were analyzed using ordinary one-way ANOVA with Bonferroni's test for multiple comparisons ($^*p < 0.05$ $^{**}p < 0.01$ $^{***}p < 0.001$). **b** MA plots depicting differential gene expression analysis of RNA-seq data obtained from transfection of TR-siRNAs and a *Tmprss6*-targeting siRNA control in Hep3B cells (top; 10 nM dose harvested at 24 h) and primary mouse hepatocytes (bottom; 50 nM dose harvested at 48 h) ($n = 4$ biological replicates per condition). Dots represent individual

mRNAs, average normalized read counts across replicates, and log$_2$ fold change relative to no siRNA controls. 'Red'-colored dots represent genes with significant differential expression (false discovery rate <0.05) but no canonical seed-matched sites within their 3' UTRs (8-mer, 7mer-m8, and 7mer-A1). 'Blue'-colored dots indicate genes with significant differential expression and canonical seed-matched sites within their 3' UTRs. 'Dark gray' dots denote genes that contain canonical 3' UTR seed-matched binding sites but are not significantly differentially expressed. The circled dot represents on-target knockdown. **c** Rat toxicity evaluation of TR-siRNAs by measurement of serum alanine aminotransferase (ALT) and alkaline phosphatase (ALP) at necropsy (24 h after last dose of siRNA). $n = 4$ males (6–8 weeks old) per group; qw weekly dosing. Statistical analysis was performed by ordinary one-way ANOVA [ALT: $F_{(6, 21)} = 0.79$, $p = 0.59$; ALP: $F_{(6, 21)} = 1.46$, $p = 0.24$] followed by Dunnett's multiple comparisons test ($p > 0.1$ for all doses compared to saline control). All error bars represent s.e.m. Source data are provided as a Source Data file.

participants[2,40]. One could envision using an RNAi-based approach to titrate transgene expression to within the therapeutic window and thereby mitigate potential negative consequences associated with over-production of the therapeutic protein, such as increased thrombotic risk in the case of high FVIII levels.

In addition to initial dose titration after AAV dosing, RNAi-based approaches could enable transgene expression to be dynamically modified as the disease state evolves. Incorporation of an RNAi-based safety switch could allow temporary cessation of treatment if needed. With vectorized shRNA co-expression, the potential to delay stable transgene expression until after immune responses to the AAV vector subside represents another potential key benefit[41]. Interestingly, with vectorized shRNA co-expression, our longitudinal in vitro studies showed that transgene expression from both the self- and non-self-silencing vectors was comparable at early timepoints, with RNAi onset occurring at a later timepoint. (Fig. 1c). The delayed onset of RNAi effects may be attributed to slower kinetics of shRNA processing to achieve a sufficient concentration of RISC-loaded siRNA for knockdown, versus the relatively faster nuclear export, and translation of the mature transgene mRNA. If these kinetic differences are recapitulated at early timepoints post-AAV administration in vivo, such a feature could find utility in approaches where only transient transgene expression is needed upon AAV dosing.

A key piece of data that emerged from our studies was that REVERSIR treatment afforded complete recovery of the full range of transgene expression that was observed in the absence of the RNAi regulatory module. A previously reported RNAi on-switch design based on ligand-promoted occlusion of a microRNA target site with a competing complementary strand yielded a maximal recovery of ~20% of non-regulated expression, likely owing to the remarkable efficiency with which RISC recognizes and binds its targets[22,42]. Our observations indicate that blocking RNAi activity directly at the level of catalytic RISC provides a significantly more efficient and robust means of regulating the on-state.

Whereas our approach facilitated maximal recovery of transgene expression, the overall dynamic range for induction was still relatively modest, with ~5–10 fold regulation of reporter gene expression detected in vivo. However, in line with previous studies showing that more efficient RNAi-mediated silencing results in wider dynamic ranges for induction, we observed the most significant upregulation with AAV-expressed hANGPTL3 where we were able to achieve >90% suppression of basal transgene expression with siRNA[22].

On the other hand, on-state control via regulation of alternative splicing with small molecules or ribozyme activity with steric oligos achieve a much larger magnitude (>100-fold) but a relatively transient duration of transgene induction[36,37]. As a result, chronic administration of the inducer might be required to maintain persistent expression of the therapeutic transgene, particularly for proteins and peptides with a short half-life. Chronic administration of REVERSIR would also be required for sustained transgene expression in our approach. However, the prolonged duration of activity resulting from a single dose of REVERSIR may support infrequent dosing. We showed that a single subcutaneous administration of REVERSIR produced weeks-long elevations in GLuc reporter, which has an extremely short serum half-life of 20 min. We also demonstrated that duration of induction can be controlled by modifying the metabolic stability of the REVERSIR, thereby adding a layer of versatility for different gene therapy applications. Additional refinements to the metabolic stability and dose of the REVERSIR could feasibly shorten timescales of induction even further.

We also report potent siRNA sequences with high specificity that can be incorporated into episomal vectors to selectively regulate exogenous transgene expression. Several studies have shown that hybridization of the siRNA seed region to complementary sequences within the 3′ UTR of mature transcripts is the principal driver of RNAi-mediated off-target gene dysregulation and in vivo toxicity[43,44]. Therefore, we prioritized siRNAs with seed-matched target sequences that occur at low frequency and whose full-length sequences lack homology to expressed transcripts across mouse, rat, cynomolgus monkey, and human genomes. We saw minimal to no seed- or non-seed-mediated dysregulation of endogenous mRNA expression in both human and mouse hepatic cell lines at high doses. All three TR-siRNAs demonstrated a favorable safety profile in a repeat-dose rat toxicity study. These data suggest that these siRNAs may be administered at relatively high doses if needed with minimal risk of the off-target effects. Overall, these features increase their potential for clinical translation with favorable safety profile. Our current studies utilized a regulatory element consisting of a single fully-matched binding site within the 3′ UTR of the AAV transgene. The sensitivity of the RNAi-driven regulation could be further influenced by modifying the local sequence around the target site or altering the proximity of the target site to the transgene stop codon[45].

While our current investigations are limited to control of hepatocyte-directed transgene expression with GalNAc-conjugated siRNAs and REVERSIR, novel delivery solutions such as C16 may broaden the scope of RNAi-based strategies for control of AAV vectors targeting a wide range of tissues[14]. For extrahepatic applications, constitutive basal transgene silencing via vectorized RNAi delivery might be preferable since it obviates the need for exogenous siRNA and enables a single agent strategy involving dosing of REVERSIR alone.

In summary, our data suggest that an RNAi-based rheostat could be a potent and clinically adaptable tool for expression control from AAV vectors, with a unique profile and use case compared to other transgene regulatory modalities.

## Methods

### Plasmids
On- and off-target reporters were generated by sub-cloning a DNA fragment containing a fully complementary (23-mer) or partial siRNA target sites into the psiCHECK2 vector (Promega C8021) between *Xho1* and *Not1* restriction sites. The on-target reporter plasmids contain a single site with full complementarity to the antisense strand cloned into the 3′-UTR of Renilla luciferase. Off-target reporter plasmids contain either one or four tandem seed-matched sites, complementary to antisense positions 2 to 9, separated by a 19-nucleotide spacer (5′-TAATATTACATAAATAAAA-3′) cloned into the 3′-UTR of Renilla luciferase[46,47].

AAV vectors were generated using Vector Builder's or Blue Heron's single-stranded AAV vector backbone with a TBG promoter, KOZAK sequence preceding the transgene, and bGH-poly A signal. For vectors harboring pri-miRNA-adapted shRNAs, the chimeric intron cassette from *pAAVsc-CB-PI-GLuc* was sub-cloned between the promoter and transgene elements, and the shRNA fragment was cloned into the PpuM1 site. The miR-33 and miR-30E shRNA sequences were designed as described previously[24,26]. A perfectly complementary siRNA target site was inserted immediately downstream of the GLuc and EPO transgenes, without any intervening spacer sequence. AAV vectors used in Fig. 2 expressed a mono- or bicistronic transcript with a fully complementary siRNA target site after the transgene stop codon, separated by a NotI restriction site. Luciferase reporter and vectorized shRNA insert sequences are detailed in Supplementary Data 1.

### Care and use of laboratory animals
All procedures and protocols performed on rodents were approved by Alnylam's Institutional Animal Care and Use Committee (IACUC) and were compliant with guidelines set by local, state, and federal regulations. Female C57BL/J mice and male Sprague Dawley rats between 6–8 weeks of age were obtained from Charles River Laboratories and allowed to acclimate in-house for 48 h prior to initiation of studies.

Animals were housed in a temperature-controlled environment maintained at 20–26 °C and 30–70% humidity under standard 12:12 h light/dark cycles. Animals were group housed (up to 5 per cage for mice and 2 per cage for rats) and provided access to standard chow and water *ad libitum*.

### AAV injections, serum and terminal tissue collections in mice

Single-stranded AAV8 vectors were generated, purified, and titered either at University of Massachusetts Viral Vector Core or Signagen Laboratories. Viral stocks were diluted in sterile 1X PBS and administered intravenously by tail vein injection at the indicated titer in a total volume of 100 µL (amount of virus injected is reported in the figure legends). Two weeks after AAV administration, mice were subcutaneously injected with GalNAc-conjugated siRNA or GalNAc-conjugated REVERSIR, or phosphate-buffered saline (PBS) as control, in a total dosing volume of 10 µL/g. All siRNA and REVERSIR test articles were diluted to the appropriate dosing concentration in 1X PBS.

Blood was collected by alternating retro-orbital bleeding under isoflurane anesthesia in accordance with IACUC approved protocols. For serum samples, blood was collected in Becton Dickinson serum separator tubes (Fisher Scientific, BD365967), kept at room temperature for 1 h and then spun in a micro-centrifuge at 21000 × $g$ at room temperature for 10 min. For plasma samples, blood was collected in Becton Dickinson plasma (K2EDTA) separator tubes (Fisher Scientific, BD365974), kept on ice for 30 min before being centrifuged at 10000 × $g$ at 4 °C for 10 min. Both serum and plasma samples were aliquoted and transferred to 96-well plates for storage at −80 °C. Animals were sacrificed at the indicated days, after which terminal livers were harvested, flash frozen in liquid nitrogen, and stored at −80 °C until further downstream analysis.

### Rat toxicity studies

Test articles were diluted with 0.9% NaCl to achieve appropriate dosing concentrations and dosed subcutaneously on the upper back to male Sprague Dawley rats in a dose volume of 5 mL/kg with $N = 4$ animals/group. Rats were sacrificed and livers and blood collected on Day 16. Randomization was performed using the partitioning algorithm in the Pristima® Suite (Xybion) that avoids group mean body weight bias. Investigators were not blinded to the group allocation during the experiment or when assessing the outcome.

Whole-venous blood was collected into serum separator tubes (BD Microtainer) and allowed to clot at room temperature for 30 min prior to centrifugation at 3000 RPM (1489 × $g$) for 10 min at 4 °C. Serum was then aliquoted and stored at −80 °C until analyses. Serum chemistries were analyzed using the AU400 chemistry analyzer (Beckman Coulter- Brea, CA, USA), with reagents provided by Beckman Coulter, Randox, and Sekisui Diagnostics.

### Oligonucleotide synthesis

Larger scale oligonucleotides were synthesized on a MerMade-12 DNA/RNA synthesizer at scales of 50–200 µmol[14,29,48]. Sterling solvents/reagents from Glen Research, 500-Å controlled pore glass (CPG) solid supports from Prime Synthesis, 2'-deoxy 3'-phosphoramidites from Thermo, and 2'-OMe and 2'-F nucleoside and LNA 3'-phosphoramidites from Hongene were all used as received. GalNAc CPG support was prepared and used as previously described[49]. Low-water content acetonitrile was purchased from EMD Chemicals. A solution of 0.6 M 5-(S-ethylthio)−1H-tetrazole in acetonitrile was used as the activator. The phosphoramidite solutions were 0.15 M in anhydrous acetonitrile with 15% DMF as a co-solvent for 2'-OMe uridine and cytidine. The oxidizing reagent was 0.02 M I2 in THF/pyridine/water. N,N-Dimethyl-N'-(3-thioxo-3H-1,2,4-dithiazol-5-yl)methanimidamide (DDTT), 0.09 M in pyridine, was used as the sulfurizing reagent. The detritylation reagent was 3% dichloroacetic acid (DCA) in dichloromethane (DCM).

After completion of the solid-phase synthesis, the CPG solid support was washed with 5% (v/v) piperidine in anhydrous acetonitrile three times with 5 min holds after each flow. The support was then washed with anhydrous acetonitrile and dried with argon. The oligonucleotides were then incubated with 28–30% (w/v) NH4OH, at 35 °C for 20 h. The solvent was collected by filtration, and the support was rinsed with water prior to analysis. Oligonucleotide solutions of approximately 1 OD260 units/mL were used for analysis of the crudes, and 30–50 µL of solution were injected. LC/ESI-MS was performed on an Agilent 6130 single quadrupole LC/MS system using an XBridge C8 column (2.1 × 50 mm, 2.5 µm) at 60 °C. Buffer A consisted of 200 mM 1,1,1,3,3,3-hexafluoro-2-propanol and 16.3 mM triethylamine in water, and buffer B was 100% methanol. A gradient from 0 to 40% of buffer B over 10 min followed by washing and recalibration at a flow rate of 0.70 mL/min. The column temperature was 75 °C. All oligonucleotides were purified and desalted, using previously reported methods[49].

### Bioinformatic prediction of transgene regulator siRNA sequences

Selection of the molecular switch duplexes was informed by conventions used for therapeutic siRNA development candidates by Alnylam: 21 nucleotide sense strand, and a 23 nucleotide antisense strand, forming a 2 base overlap at the 3'-end of the guide strand.

A set of all decamers was generated in silico to represent the first 10 bases of a candidate antisense (guide) sequence (1,048,576 sequences). miRNA seed sequences were retrieved from the miRbase database for *Homo sapiens*, *Mus musculus*, *Rattus norvegicus*, *Macaca mulatta* and *Macaca nemestris*. The non-human primate species were selected as proxies for *Macaca fascicularis* (cynomolgus monkey), which is not represented in miRbase. Decamers containing seed sequences (bases 2-7) matching the miRNA seed sequences were removed from further consideration (487,186 decamers remaining).

Each remaining decamer was annotated with a predicted quiescence score based on a proprietary off-target prediction algorithm. Those with scores in the lowest quartile (predicted most quiescent) were retained (155,374 decamers), and others removed from consideration.

The frequency of heptamers found in the human transcriptome (NCBI RefSeq), as was computed by aligning transcripts for each gene, and counting all heptamer substrings in the global alignment consensus and non-overlapping regions. Decamers were then annotated with the frequency of the heptamer in the seed (positions 2-8), and those in the lowest decile heptamer frequency were retained (1767 decamers) and the remainder removed from consideration.

The remaining decamers were then aligned to the human, mouse, rat, and cynomolgus monkey transcriptomes using BLASTN with the parameters "-task blastn-short -dust no -evalue 1000 -ungapped -perc_identity 100" to count the number of occurrences of each decamer within each transcriptome. The decamers were then sorted in ascending order based on the total number of identical alignments on the reverse complement strand, then predicted quiescence, and number of seed matches in the human transcriptome.

For each decamer, 10,000 random 13-mer sequences were created and appended to create 10,000 candidate 23-mer siRNA guide sequences with a common 10-mer prefix. Each 23-mer was then aligned to the transcriptomes of human, mouse, rat, and cynomolgous monkey using a weighted ungapped alignment of the 23-mer to the transcripts, (mismatch penalty for positions 2-9 is 2.8, for positions 10-11, 1.2, for positions 12–19, 1.0, and for positions 1 and 20–23, 0.0). For each candidate decamer prefix, the 23-mers with the top 10 worst alignment profiles (most poorly aligned to the transcriptomes) were retained. Sequences were sorted by their alignment score and predicted quiescence, and top candidates were selected.

## ELISA assays

Circulating AAV-expressed human ANGPTL3 protein levels were measured from plasma using commercially-available ELISA kits (plasma diluted 1:4 and used with R&D Systems #DANL30). The assay is specific for detection of human ANGPTL3 protein, with no significant cross-reactivity to other related angiopoietin molecules or mouse ANGPTL3. Mouse EPO concentrations were measured with Mouse EPO Quantikine ELISA kit from R&D Systems MEP00B (serum diluted 1:150). All assays were performed following the manufacturer's protocols.

## Cell lines and transfection

Cos-7 (ATCC CRL-1651) and HepG2 (ATCC HB-8065) cells were grown in DMEM and EMEM, respectively, both supplemented with 10% heat-inactivated FBS and 1% glutamine and maintained in a humidified incubator at 37 °C, 5% $CO_2$. Plasmids and siRNAs were co-delivered by reverse transfection using Lipofectamine 2000 (Thermo Fisher Scientific 11668) for Cos-7 cells and Lipofectamine 3000 for HepG2 cells, following the manufacturer's protocol.

## Luciferase reporter assays

*siRNA on-target and off-target reporter evaluations:* Cos7 cells were co-transfected with 5 ng psiCHECK2 reporter plasmid and the specified amounts of siRNA duplexes (serially diluted in PBS) in a 384-well plate format at a density of $5 \times 10^3$ cells per well. To assess REVERSIR-mediated rescue of knockdown by vector-encoded shRNAs expressed in trans, Cos7 cells were co-transfected in the same 384-well format with 70 ng of shRNA expression plasmid, 5 ng of psiCHECK2 reporter, in addition to indicated amounts of the specified REVERSIR molecules (serially diluted in PBS). 48 h post-transfection, Firefly (transfection control) and Renilla (target) luciferase activities were sequentially measured using the Dual-Glo Luciferase Assay System (Promega E2920) and detected on a Spectramax M plate reader (Molecular Devices). The Renilla signal was normalized to Firefly signal for each well and expressed as a percentage relative to control wells transfected with reporter alone without siRNA or non-targeting shRNA plasmid.

In vitro *characterization of self-silencing AAV constructs:* HepG2 cells were seeded at a density of $2 \times 10^4$ cells per well in a 96-well plate and co-transfected with 16.6 ng intronic shRNA-containing GLuc expression plasmid along with the indicated dose of REVERSIR. As an internal control to normalize for transfection efficiency, 3.3 ng of PGK-driven Luc2 (pGL4.53[luc2/PGK]) vector was also co-transfected, constituting 17% of the total transfected DNA. Reverse transfections were carried out with 0.1 µL P3000 and 0.2 µL Lipofectamine 3000 per well and allowed to proceed for 6 h after which the media was replaced. Expression of GLuc and Luc2 reporters was measured 48 h after transfection. To measure secreted GLuc levels, cell culture supernatant from each sample was diluted 1:50 in EMEM. 5 µL of diluted supernatant and 50 µL of assay buffer containing 3 µM coelenterazine substrate (Selleck Chem S7777; stocks made up to 1 mM in DMSO and subsequently diluted to 3 µM in IX PBS) were transferred to each well of a white opaque 96-well plate and read on a Spectramax L microplate luminometer.

Plate was dark-adapted to minimize auto-luminescence and injection speed was set to 250 µL/s, followed by 2 s shake and 1 s signal integration time per well. To determine cellular Luc2 expression, cells in each well were first lysed with 50 µL ice-cold 1X passive lysis buffer (Promega E1941), allowed to incubate on an orbital shaker for 10 min at room temperature, after which 50 µL of ONE-Glo EX luciferase reagent (Promega E8110) was added. Luciferase signals were detected on a SpectraMax M plate reader 5 min later. Ratio of GLuc to Luc2 intensity was computed for each well.

*Longitudinal monitoring of secreted GLuc levels* in vivo: 5 µL of serum was assayed as described above on a Spectramax L luminometer. All serum samples corresponding to a study were run simultaneously under uniform conditions and resulting GLuc signals at an individual timepoint were expressed as percent relative to that at D0 (pre-treatment with siRNA or REVERSIR) for each animal.

## RNA isolation and RT-qPCR evaluations

96-well plates of cells were directly homogenized in 100 µL RLT buffer (Qiagen) containing 10 µL/mL beta-mercaptoethanol and allowed to incubate on an orbital shaker for 10 min at room temperature. RNA was extracted using the Qiagen RNeasy 96 kit (#74181), with additional incorporation of an on-column DNase digestion step (Qiagen 79254). For in vivo samples, powdered liver (~10 mg) was resuspended in 900 µL QIAzol (RNeasy 96 Universal Tissue Kit, Qiagen, 74881) and homogenized at 25 cycles per second for 1 min at 4 °C using a TissueLyser II (Qiagen, 85300). Samples were incubated at room temperature for 5 min followed by addition of 180 µl chloroform. Samples were vigorously mixed, followed by a 10 min incubation at room temperature. Samples were spun at $12000 \times g$ for 15 min at 4 °C, the supernatant was moved to a new tube, and 1.5 volumes of 70% ethanol was added. Samples were then purified using a RNeasy 96 Universal Tissue Kit (Qiagen, 74881) with on-column DNase digestion. RNA was eluted from the RNeasy spin columns with 50 µl RNAse-free water (Ambion) and quantified on a Nanodrop (Thermo Fisher Scientific). 0.5 µg of purified RNA was reverse transcribed using either the iScript gDNA Clear cDNA Synthesis Kit (BioRad, #1725034) or Maxima H Minus First Stand cDNA Synthesis Kit (Thermo Fisher Scientific M1681). RNA was DNase-treated prior to cDNA synthesis and oligo(dt)$_{18}$ primers were use, where applicable. The product was diluted 1:2 in RNase-free water and subjected to quantitative real-time PCR (qRT-PCR) using gene-specific TaqMan assays (Thermo Fisher Scientific 4331182) for mouse TTR (Mm00443267_m1), human PMP22 (Hs00165556_m1), Luc2 (forward primer: TAAGGTGGTGGACTTGGACA, reverse primer: GTTGTTAACGTAGCCGCTCA, FAM-MGB probe: CGCGCTGGTTCACACCCAGT), and GLuc (Catalog #4441114 ARKA6G2). Levels of mouse (Mm99999915_g1) or human (Hs99999905_m1) GAPDH were used as endogenous normalization controls. Real-Time PCR was performed in a Roche LightCycler 480 using LightCycler 480 Probes Master Mix (Roche, 04707494001). No-reverse transcriptase enzyme controls were performed on a few samples in each study to ensure lack of viral or genomic DNA contamination. All RT-qPCR data were analyzed using the ΔΔCt method.

## RNA-seq methods

Primary Mouse Hepatocytes (BIOIVT, Cat # M005052-P, Lot GBW) were transfected in 384-well plates (5000 cells per well) with siRNA or DPBS (mock control) at a final concentration of 50 nM using Invitrogen Lipofectamine® RNAiMAX (Invitrogen, Carlsbad, CA, Catalog No. 13778-150). After 48 h, cell were lysed in lysis buffer (Tris HCl pH 7.5 100 mM, LiCl 500 mM, EDTA pH 8.0 10 mM, LiDS 1%, DTT 5 mM supplemented with TURBO™ DNase, Thermofisher #AM2238) for 30 min. Subsequently, whole transcriptome mRNA-enriched RNA-Seq libraries were constructed from cell lysates using a KAPA mRNA Capture Kit and mRNA HyperPrep Kit (Roche) as per the manufacturer's protocol. RNA-Seq libraries were quantified by low-depth sequencing on Illumina iSeq instrument. Equal amounts of each library/sample were pooled and sequenced on an Illumina NovaSeq instrument with 2 x 150bp paired-end settings, according to manufacturer's instructions. Hep3B cells (ATCC HB-8064) were transfected in the same format at 10 nM concentration and harvested 24 h post-transfection.

Raw RNAseq reads were filtered with minimal mean quality scores of 28 and minimal remaining length of 36, using the ea-utils software fastq-mcf v1.05 (https://expressionanalysis.github.io/ea-utils/). Filtered reads were aligned to the *Mus musculus* genome (GRCm39/mm39) or *Homo sapiens* genome (GRCh38.p13) using STAR (ultrafast universal RNAseq aligner) v2.7.9a[50]. Uniquely aligned reads were counted by featureCounts v2.0.2 with the minimum mapping quality score set to 10[51]. Differential gene expression analysis was performed

by the R package DESeq2 v1.34.0 with the betaPrior parameter set to TRUE to shrink $\log_2$ fold-change estimates for noisy, low-count genes[52].

## Statistical analysis

Statistical analyses were performed using GraphPad Prism v.7 software. Data were analyzed by one-way or two-way analyses of variance (ANOVA) followed by Tukey's, Bonferroni's, Dunnett's, or Holm-Sidak's post hoc tests for multiple comparisons. In certain instances where indicated, individual unpaired two-tailed *t*-test or multiple *t*-testing (one per row) with Holm-Sidak correction was employed. All data are presented as mean +/- s.e.m. and differences between groups considered significant if ***$p < 0.0001$, ***$p < 0.001$, **$p < 0.01$, or *$p < 0.05$. Relevant non-significant differences between groups are highlighted with "n.s."

## Reporting summary

Further information on research design is available in the Nature Portfolio Reporting Summary linked to this article.

## Data availability

The data supporting the findings of this study are available from the corresponding authors upon reasonable request. The raw RNA sequencing data reported in this manuscript have been deposited to the NCBI Gene Expression Omnibus and are accessible through GEO Series accession number GSE214065. Source data are provided with this paper.

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

## Acknowledgements

The authors thank Alnylam's in vivo Sciences Team for conducting the safety studies and the Pathology team for generating the clinical pathology data. Schematics were created with BioRender.com.

## Author contributions

Conceived studies: M.S., J.M., C.K., M.M., K.F., V.J. Designed studies and experiments: M.S., J.M., I.Z., M.K.S., C.K., S.A., V.J. Performed experiments: M.S., J.M., C.K., T.N., S.A., T.R., M.A.A., K.W., Ty.C., Th.C., E.S., S.S.M., A.B. Analyzed and interpreted data: M.S., K.B., D.B., J.B., S.A., J.M., I.Z., C.K., C.R.B., A.B.C., K.S., V.J. Supervised: M.M., K.F., V.J. Prepared the manuscript with input from all authors: M.S., V.J.

## Competing interests
