## [Peer Review File · Nature Communications]

RNAi-mediated Rheostat for Dynamic Control of AAV-Delivered TransgenesREVIEWERS' COMMENTS

Reviewer #1 (Remarks to the Author):

This is a very impressive study. One problem with gene therapy is that gene expression might be too high and too uncontrollable. This paper describes the use of an shRNA or siRNA to down-regulate expression of the critical gene and a related ASO to block the shRNA to allow normal regulation. Because the siRNA or ASO are synthetic, they can be added to "tune" expression as desired. One could imagine a clinician monitoring the expression of a gene therapy and tuning it up or down with a "kit" of synthetic ASOs and duplex RNAs. These impressive results are in line with the know state-of-the-art capabilities of ASOs and duplex RNAs, so, in a sense, the results are not surprising. What is gratifying is the demonstration that a system with so many moving parts works as intended. This is synthetic biology reduced to practice in a highly professional manner. Indeed, I am not aware of any other paper that shows this level of complexity in tuning, making it an excellent fit for Nature Communications. The quality of the data are excellent. Controls are appropriate. The demonstration includes experiments in both cell culture and animals. Multiple different targets are addressed to show generality.

The text is well written, except that the paragraphs are overly long and complex. Simply breaking up the paragraphs so that they only deal with one subject would be helpful. Similarly, the Figures are too long. For example, what is the purpose of including both cell culture and animal data in Figure 1. Does Nature Communications have a limit on the number of Figures? What is the benefit to having a complex multi-part figure that requires a long, complex Figure legend to be on a separate page. Finally, if the format allows, I recommend adding sub-headings to the introduction.

One request. Could the authors comment on the implications of designing and getting approval for a trial that might require three agents (gene therapy, duplex RNA, ASO). The theoretical advantages are made clear in the text, but is regulatory permission to try this realistic? Any honest answer to this question will be welcome, whether positive or negative for the future of the technology.

In summary, this is one of the best papers I have read in a long time. Quality team effort that thoroughly addresses the topic.

Reviewer #2 (Remarks to the Author):

Subramanian et al described a method to fine-tune recombinant adeno-associated virus (rAAV)-mediated liver gene expression. The method builds upon two established technologies: RNA interference (RNAi) that silences gene expression and REVERSIR, a synthetic oligonucleotide that can quench RNAi and therefore restore gene expression. Precise control of the timing and transgene expression level is much needed in the gene therapy field, which may mitigate toxicity associated with uncontrolled transgene expression. The method described in this study may provide a solution to a better and safer gene therapy. The data well support the claim, and the manuscript is well written.

This reviewer has only minor suggestions:

1. Page 20, lines 418-419. "..., the transient expression of transgene is seen before the RNAi effect is observed" is an interesting finding and relevant data should be presented and referenced.
2. Although the method is conceptually applicable to various tissue/cell types, the study is limited to hepatocyte-directed gene therapy mainly due to easy delivery. I appreciate the relevant discussion at the end of the manuscript (lines 457-462), but would suggest the authors clarify the hepatocyte-directed scope of the study in the abstract. After all, extrahepatic delivery of siRNA and oligonucleotide is not an easy task despite recent advances.

Reviewer #3 (Remarks to the Author):

In this manuscript Subramanian and colleagues developed a molecular switch to regulate transgene expression in the context of AAV-mediated gene transfer. The work appeared to be divided in two parts. The first is aimed at investigating the efficacy of REVERSIR technology, consisting of oligonucleotides complementary to the siRNA guide strand, to reverse siRNA- and shRNA-mediated RNAi activity on AAV-delivered transgenes. In the second part, the authors developed novel synthetic siRNA sequences to regulate transgene expression. Although addressing a common topic, the two parts of the study appeared to be unrelated and to include a relatively small amount of data. From a methodological point of view, studies are well done and results are solid and well presented.

Tuning of transgene expression is emerging as an important issue in the context of clinical gene therapy. However, application of REVERSIR compounds to AAV gene therapy is rather attractive because it would imply the administration of three different medicinal products (AAV, siRNA, and REVERSIR compound) to the same patient. The use of self-silencing AAVs, including shRNA in the expression cassette, would address this issue as it requires the administration of only two components. Nevertheless, induction of transgene expression from such a vector would require chronic administration of REVERSIR compound. Moreover, as also discussed by the authors, the overall dynamic range of REVERSIR-mediated modulation on gene expression is modest.

Overall, this reviewer is not convinced about the relevance of the REVERSIR technology to the AAV gene transfer and the paper lacks a proof-of-efficacy experiment in a disease model, clearly showing the advantages brought by this technology to the field. Finally, the work is somehow additive to previous works from these authors and does not contain major science innovations.

NCOMMS-22-46350-T
Response to Reviewers

Reviewer #1 (Remarks to the Author):

This is a very impressive study. One problem with gene therapy is that gene expression might be too high and too uncontrollable. This paper describes the use of an shRNA or siRNA to down-regulate expression of the critical gene and a related ASO to block the shRNA to allow normal regulation. Because the siRNA or ASO are synthetic, they can be added to "tune" expression as desired. One could imagine a clinician monitoring the expression of a gene therapy and tuning it up or down with a "kit" of synthetic ASOs and duplex RNAs. These impressive results are in line with the know state-of-the-art capabilities of ASOs and duplex RNAs, so, in a sense, the results are not surprising. What is gratifying is the demonstration that a system with so many moving parts works as intended. This is synthetic biology reduced to practice in a highly professional manner. Indeed, I am not aware of any other paper that shows this level of complexity in tuning, making it an excellent fit for Nature Communications. The quality of the data are excellent. Controls are appropriate. The demonstration includes experiments in both cell culture and animals. Multiple different targets are addressed to show generality.

The authors appreciate the thorough review and the valuable suggestions and have addressed the specific points raised by in the responses below and as appropriate, in the respective sections of the manuscript.

The text is well written, except that the paragraphs are overly long and complex. Simply breaking up the paragraphs so that they only deal with one subject would be helpful.

We simplified by breaking up long paragraphs. We also simplified the text by breaking up overly long sentences.

Similarly, the Figures are too long. For example, what is the purpose of including both cell culture and animal data in Figure 1. Does Nature Communications have a limit on the number of Figures? What is the benefit to having a complex multi-part figure that requires a long, complex Figure legend to be on a separate page.

We split up Figure 1 into Figures 1 and 2. Figure 1 shows in vitro and in vivo data using the GLuc reporter vector. Figure 2 shows in vivo data using the physiologically relevant EPO transgene.

Finally, if the format allows, I recommend adding sub-headings to the introduction.

The format may not allow addition of sub-headings to the introduction. However, we simplified the text by breaking up long sentences and paragraphs.

One request. Could the authors comment on the implications of designing and getting approval for a trial that might require three agents (gene therapy, duplex RNA, ASO). The theoretical advantages are made clear in the text, but is regulatory permission to try this realistic? Any honest answer to this question will be welcome, whether positive or negative for the future of the technology.

The authors agree that is a valid and important consideration. There is wealth of data now with GalNAc-siRNA conjugates for multiple targets. This adds confidence for getting the human safety data for self-regulating siRNAs. We do acknowledge there is no human data so far with Reversir oligos and that would

need to be generated. After getting the human safety data for these components, we do see the path for using them in a trial.

In summary, this is one of the best papers I have read in a long time. Quality team effort that thoroughly addresses the topic.

The authors appreciate the overall positive response of the reviewer and the helpful comments.

Reviewer #2 (Remarks to the Author):

Subramanian et al described a method to fine-tune recombinant adeno-associated virus (rAAV)-mediated liver gene expression. The method builds upon two established technologies: RNA interference (RNAi) that silences gene expression and REVERSIR, a synthetic oligonucleotide that can quench RNAi and therefore restore gene expression. Precise control of the timing and transgene expression level is much needed in the gene therapy field, which may mitigate toxicity associated with uncontrolled transgene expression. The method described in this study may provide a solution to a better and safer gene therapy. The data well support the claim, and the manuscript is well written.

The authors appreciate the thorough review and the valuable suggestions, and have addressed the specific points raised in the responses below and as appropriate, in the respective sections of the manuscript.

This reviewer has only minor suggestions:

1. Page 20, lines 418-419. "..., the transient expression of transgene is seen before the RNAi effect is observed" is an interesting finding and relevant data should be presented and referenced.

We thank the reviewer for the comment. We have added a more detailed discussion to clarify this statement and referred to the relevant figure in the manuscript: "Interestingly, with vectorized shRNA co-expression, our longitudinal *in vitro* studies showed that transgene expression from both the self- and non-self-silencing vectors was comparable at early timepoints, with RNAi onset occurring at a later timepoint. (Figure 1c). The delayed onset of RNAi effects may be attributed to slower kinetics of shRNA processing to achieve a sufficient concentration of RISC-loaded siRNA for knockdown versus the relatively faster nuclear export, and translation of the mature transgene mRNA. If these kinetic differences are recapitulated at early timepoints post-AAV administration *in vivo*, such a feature could find utility in approaches where only transient transgene expression is needed upon AAV dosing."

2. Although the method is conceptually applicable to various tissue/cell types, the study is limited to hepatocyte-directed gene therapy mainly due to easy delivery. I appreciate the relevant discussion at the end of the manuscript (lines 457-462), but would suggest the authors clarify the hepatocyte-directed scope of the study in the abstract. After all, extrahepatic delivery of siRNA and oligonucleotide is not an easy task despite recent advances.

We agree that this limitation should be addressed in the abstract and have added the words "hepatocyte-directed" to the following sentence in the abstract: "We present a generalizable RNAi-based rheostat wherein **hepatocyte-directed** AAV transgene expression is silenced using the clinically validated modality of chemically modified short interfering RNA (siRNA) conjugates or vectorized co-expression of short hairpin RNA (shRNA)."

Reviewer #3 (Remarks to the Author):

In this manuscript Subramanian and colleagues developed a molecular switch to regulate transgene expression in the context of AAV-mediated gene transfer. The work appeared to be divided in two parts. The first is aimed at investigating the efficacy of REVERSIR technology, consisting of oligonucleotides complementary to the siRNA guide strand, to reverse siRNA- and shRNA-mediated RNAi activity on AAV-delivered transgenes. In the second part, the authors developed novel synthetic siRNA sequences to regulate transgene expression. Although addressing a common topic, the two parts of the study appeared to be unrelated and to include a relatively small amount of data. From a methodological point of view, studies are well done and results are solid and well presented.

The authors appreciate the thorough review and the valuable critiques, and have addressed the specific points raised by Reviewer #3 in the responses below and as appropriate, in the respective sections of the manuscript.

Tuning of transgene expression is emerging as an important issue in the context of clinical gene therapy. However, application of REVERSIR compounds to AAV gene therapy is rather attractive because it would imply the administration of three different medicinal products (AAV, siRNA, and REVERSIR compound) to the same patient. The use of self-silencing AAVs, including shRNA in the expression cassette, would address this issue as it requires the administration of only two components.

Nevertheless, induction of transgene expression from such a vector would require chronic administration of REVERSIR compound. Moreover, as also discussed by the authors, the overall dynamic range of REVERSIR-mediated modulation on gene expression is modest.

We agree with the potential issues raised by the reviewer. We included the following statement to the discussion: "Chronic administration of REVERSIR would also be required for sustained transgene expression in our approach. However, the prolonged duration of activity resulting from a single dose of REVERSIR may support infrequent dosing."

Overall, this reviewer is not convinced about the relevance of the REVERSIR technology to the AAV gene transfer and the paper lacks a proof-of-efficacy experiment in a disease model, clearly showing the advantages brought by this technology to the field. Finally, the work is somehow additive to previous works from these authors and does not contain major science innovations.

We appreciate the feedback of reviewer and agree that the results are not surprising. But as pointed out by Reviewer 1, this work is the demonstration that a system with so many moving parts works as intended. We also acknowledge that chronic administration of REVERSIR would also be required for sustained transgene expression in our approach. However, the prolonged duration of activity resulting from a single dose of REVERSIR may support infrequent dosing.